# Immunogenicity of Oral Rabies Vaccine Strain SPBN GASGAS in Local Dogs in Bali, Indonesia

**DOI:** 10.3390/v15061405

**Published:** 2023-06-20

**Authors:** Irene Linda Megawati Saputra, Suwarno Suwarno, Wahid Fakhri Husein, Pebi Purwo Suseno, I Made Angga Prayoga, Ad Vos, I Made Arthawan, Luuk Schoonman, John Weaver, Nuryani Zainuddin

**Affiliations:** 1Directorate of Animal Health, Ministry of Agriculture, Jakarta 12550, Indonesia; drhirene88@gmail.com (I.L.M.S.); pebi212@gmail.com (P.P.S.); nuryanizainuddin@yahoo.com (N.Z.); 2Faculty of Veterinary Medicine, Airlangga University, Surabaya 60115, Indonesia; suwarno@fkh.unair.ac.id; 3Emergency Centre for Transboundary Animal Diseases, The Food and Agriculture Organization of the United Nations, Jakarta 12550, Indonesia; wahid.husein@fao.org (W.F.H.); luuk.schoonman@fao.org (L.S.); 4Australia Indonesia Health Security Partnership, Jakarta 12550, Indonesia; anggalepok23@gmail.com (I.M.A.P.); john.weaver@aihsp.or.id (J.W.); 5Veterinary Public Health, Ceva SA, 33500 Libourne, France; 6Bali Provincial Animal Health Services, Den Pasar 80225, Indonesia; keswanbali@gmail.com

**Keywords:** Bali, dog, ORV, rabies, SPBN GASGAS

## Abstract

Dog-mediated rabies is endemic in much of Indonesia, including Bali. Most dogs in Bali are free-roaming and often inaccessible for parenteral vaccination without special effort. Oral rabies vaccination (ORV) is considered a promising alternative to increase vaccination coverage in these dogs. This study assessed immunogenicity in local dogs in Bali after oral administration of the highly attenuated third-generation rabies virus vaccine strain SPBN GASGAS. Dogs received the oral rabies vaccine either directly or by being offered an egg-flavored bait that contained a vaccine-loaded sachet. The humoral immune response was then compared with two further groups of dogs: a group that received a parenteral inactivated rabies vaccine and an unvaccinated control group. The animals were bled prior to vaccination and between 27 and 32 days after vaccination. The blood samples were tested for the presence of virus-binding antibodies using ELISA. The seroconversion rate in the three groups of vaccinated dogs did not differ significantly: bait: 88.9%; direct-oral: 94.1%; parenteral: 90.9%; control: 0%. There was no significant quantitative difference in the level of antibodies between orally and parenterally vaccinated dogs. This study confirms that SPBN GASGAS is capable of inducing an adequate immune response comparable to a parenteral vaccine under field conditions in Indonesia.

## 1. Introduction

In Indonesia, rabies was first reported in West Java in the late 19th century, and in subsequent decades it was also reported from the other major islands. Rabies is now considered endemic in 26 of the country’s 38 provinces. Bali was free of rabies until an incursion in 2008 and is now endemically infected [1]. Tailored dog vaccination strategies were developed by the national and local governments in Bali, and implemented in collaboration with other international stakeholders, but the goal of a rabies-free Bali has yet to be achieved [2]. 

A major challenge to rabies control is that a high proportion of dogs are ostensibly owned but free-roaming, and most owners cannot restrain the dogs for vaccination [3,4]. The enhanced vaccination strategy sought to go door-to-door instead of vaccinating the dogs at a central point to increase the overall vaccination coverage; however, vaccination coverage was still not sufficient to interrupt virus circulation and has not been able to sustain vaccination coverage above the target 70%. The 70% threshold is considered necessary to prevent the spread of the disease among dogs [5]. Most likely, a much lower vaccination coverage could successfully interrupt the transmission cycle, but a higher vaccination coverage is targeted to compensate for the high population turnover between two successive vaccination campaigns [6]. Other improvements to increase vaccination coverage in Bali included emergency local vaccinations in response to reported rabies cases, additional ‘topping up’ vaccination between the rounds of mass dog vaccination to maintain herd immunity, and the establishment of well-trained dog-catching teams to systemically cover target areas. The free-roaming dogs that were difficult to restrain were caught using nets, and then vaccinated, marked, and released [7,8]. After the initial success with the dog-catching teams, dogs became more wary and more difficult to catch during subsequent campaigns [3].

To improve vaccination coverage, oral rabies vaccination (ORV) of difficult-to-catch dogs offers a suitable, cost-effective alternative. To administer ORV, dogs are offered an attractive bait, the bait is picked up and the animal must perforate the vaccine-loaded sachet incorporated in the bait. The vaccine must be released in the oral cavity, where it is taken up via the mucosal membrane and tonsils and subsequently induces an immune response [9]. Oral vaccination of targeted wildlife species, such as the red fox, raccoon dog, and coyote, has eliminated the disease from these reservoir species in large areas of North America and Europe [10,11]. ORV of dogs has been suggested as a complementary tool to mass parenteral dog vaccination to increase overall vaccination coverage and, specifically, to cover the free-roaming dog population, which is considered to play a key role in the transmission of rabies [9,12,13]. 

Before using ORV on a large scale, the World Health Organization (WHO) suggests using it in pilot studies on dogs to evaluate its feasibility and effectiveness [14]. Hence, feasibility studies have been carried out in Bali. The third-generation vaccine virus, SPBN GASGAS, derived from SAD L16, a cDNA clone of the oral rabies virus vaccine strain SAD B19, was incorporated in an egg-flavored bait for these studies. SPBN GASGAS lacks the pseudogene. Also, all three nucleotides were changed at amino acid positions 194 and 333 of the glycoprotein. As a result of the genetic modification at amino acid position 333 of the glycoprotein, the construct is no longer pathogenic in adult mice after intracerebral inoculation (i.c.). Site-directed mutagenesis at amino acid position 194 prevents a possible reversion to virulence. Further, the construct contains a second identical glycoprotein gene with modifications as described above. It was predicted that the overexpression of the rabies virus glycoprotein increased not only its efficacy but also its safety profile by reducing the potential risk of reversion to virulence and increasing apoptosis [15]. This vaccine strain, used for the ORV of wildlife in Europe, has been shown to be immunogenic in dogs under different settings [16,17,18,19]. Also, the egg-flavored bait has been readily accepted by dogs in different areas [20,21,22].

The goal of this study was to determine if the selected bait could release the vaccine in the oral cavity effectively for the SPBN GASGAS vaccine strain and induce an appropriate immune response in Bali’s local dogs. Further, the safety of the vaccine bait under field conditions was assessed through 30-day post-vaccination monitoring to identify any adverse reactions in the vaccinated dogs.

## 2. Materials and Methods

### 2.1. Study Design

The study was designed as a randomized controlled research study and conducted in local areas of the Bali Province, namely Nongan village, Karangasem District, both representing rural areas, and Banyuning village, Buleleng District, both representing urban areas (Figure 1). The study was conducted in April–May 2022.

The inclusion criteria for this study were that the dog should be easy to restrain, in good health (by visual inspection) and has never received a rabies vaccination (tested seronegative prior to vaccination). The dogs were fed and managed by their owners as usual. ELISA tests were used to check base level blood samples (B0) prior to vaccination if the dogs had any antibodies against rabies virus (RABV). 

First, 50 dogs were offered an egg-flavoured vaccine bait containing SPBN GASGAS (3.0 mL, 10^8.4^ FFU/mL). This highly attenuated third-generation oral RABV vaccine strain has been tested for safety and efficacy according to international requirements in many different target and non-target species [23,24,25,26]. Households with the selected dogs meeting the inclusion criteria were revisited on the day of vaccination and after obtaining written consent from the dog owners, dogs were offered a bait and bait acceptance/comsumption was observed. If the dog discarded the (perforated) sachet, this was collected by the team in order to reduce unintentional direct human contact with the vaccine virus. If animals did not accept the baits readily, 25 dogs received the same dose of SPBN GASGAS by direct oral administration (d.o.a.), and an additional 25 dogs were targeted for vaccination by the parenteral route (subcutaneous, s.c.) with a commercially available inactivated vaccine (Rabisin, Boehring Ingelheim Animal Health, 69007 Lyon, France). Ten dogs were included as a control group and did not receive any treatment.

An additional blood sample (B1) was collected from the dogs between 27 and 32 days post-vaccination (dpv), including samples from the control group (Figure 2). The health of the dogs was monitored once a week for the period between vaccination and blood sampling (B1) by visual examination during house visits. 

### 2.2. Assays 

Blood samples of at least 4 mL were collected from the large superficial veins of the extremities (e.g., *V. cephalica antebrachia*, *V. saphena*). The samples were transported to the Disease Investigation Center (DIC) in Denpasar at ambient temperature within 72 h. Serum was prepared from the clotted blood samples, divided into three aliquots and stored until analysis at ≤−15 °C.

Antibodies against the rabies virus were detected using a commercial blocking enzyme-linked immunosorbent assay kit (BioPro Rabies ELISA, O.K. Servis BioPro, Prague -Czech Republic). The study used this ELISA because it has been shown to provide reliable results and the obtained results from this study could be compared directly with previous studies using the same ELISA [27,28,29]. In brief, serum samples were incubated on microtiter plates coated with rabies antigen. After removing the sera, all wells were incubated with a fixed amount of biotin-labelled rabies-specific antibody, followed by incubation of the bound antibody with peroxidase-conjugated streptavidin and, then, chromophoric detection. A percentage of blocking (PB) lower than 40% was considered negative; a PB equal to or higher than 40% was considered positive.

### 2.3. Statistical Analysis

Univariate contingency table testing (Fisher’s exact test) and multiple logistic regression (MLR), if appropriate, were identified as the first two steps required in the statistical analysis. The dependent variable was “seropositivity” (yes/no), as defined by the cut-off value (PB ≥ 40%). Independent variables were the study area (Karangasem, Buleleng), level of supervision (restricted, free-roaming), size of dog (small (<10 kg), medium (10–30 kg), large (>30 kg)), sex (female, male), and age (juvenile (<12 months), adult (≥12 months)). Treatment effect (bait, d.o.a., s.c., control) was examined using an ANOVA. Variables with *p* ≤ 0.20 (univariate analysis) were to be included in the final MLR model. Statistical analyses were carried out using GraphPad Prism v9.0 (GraphPad Prism Software Inc., San Diego, CA, USA).

## 3. Results

A total of 202 local dogs were identified to be included from both study sites: Karangasem—102 dogs, and Buleleng—100 dogs. Twenty-seven dogs were excluded for different reasons; some dogs were lost or had died, other dogs were very aggressive. The remaining 175 dogs were selected for the collection of the pre-vaccination blood sample (B0). The ELISA for B0 showed 145 dogs tested negative for rabies antibodies and 30 dogs tested positive. From the total of 145 seronegative dogs, 105 were included in this study, and the remaining 40 were excluded but still received parenteral vaccination. 

Originally, it was planned to include 50 dogs in the d.o.a. group. However, one of these dogs snatched away a bag with five baits and ran off. Later on, the empty bag was found, and it was assumed that the dog also ate the five additional baits. Hence, the sample size had to be adjusted to 45 as no surplus baits were available.

A total of 13 dogs died between vaccination and the second blood sample, 10 from Karangasem and three from Buleleng: nine died from a canine parvovirus infection based on clinical signs (one from the control group), two were hit by a car, one was poisoned, and one from the control group was infected with rabies as confirmed by the presence of rabies antigen in the brain (FAT). The other dogs remained healthy during the observation period, except for one dog offered a bait, that was reported sick in the week following vaccination. This dog recovered and was reported healthy during the following three weeks. The animal that likely consumed multiple baits also remained healthy during the observation period.

The seroconversion rate of the dogs vaccinated is shown in Table 1. None of the control animals seroconverted, and in the other treatment groups, 90.5% of the animals developed a detectable immune response. Five and two dogs in the bait and s.c. groups, respectively, did not seroconvert. Also, one d.o.a. dog did not develop detectable antibodies against RABV. There was no significant difference in seroconversion rate between dogs receiving a vaccine bait or by d.o.a.: Fisher’s exact test, *p* > 0.99. Consequently, these two groups were pooled (oral) and compared with the s.c. group: No significant difference in seroconversion rate was observed: Fisher’s exact test, *p* > 0.99.

A quantitative analysis of the measured immune response post vaccination showed that only the control group deviated significantly from the other treatment groups (Figure 3) by ANOVA with Turkey’s multiple comparison test. No quantitative difference was found between the three vaccination groups. The univariate analysis of the other variables did not detect any significant effect: study area (*p* ≥ 0.99), size of dog (*p* = 0.44), sex (*p* ≥ 0.99), age (*p* = 0.71), and level of restriction (*p* = 0.71), Fisher’s exact test. Therefore, no MLR analysis was performed. 

## 4. Discussion

The results of the rabies ELISA test showed that from 45 serum samples collected from dogs treated with vaccine baits, 40 (88.9%) were seropositive and the remaining 5 (11.1%) were seronegative. Meanwhile, for the dogs that received the vaccine d.o.a., 16 from a total of 17 samples showed seropositive results (94.1%), and the remaining sample (5.8%) was seronegative. Similar results also occurred in the sample of dogs given parenteral vaccine treatment: from 22 samples, 20 (90.1%) were seropositive and 2 (9.1%) were seronegative. The dogs in the control group were all seronegative until the end of the study. The lack of seroconversion in some of the orally vaccinated animals does not necessarily indicate vaccine failures, as many factors can contribute to the absence of antibodies in these so-called non-responders [30]. In case of ORV, timely release is essential for a successful vaccination attempt. The vaccine, based on a replication-competent rabies virus, will lose its immunogenic potential rapidly once entering the gastro-intestinal tract. Many factors can affect the release of the vaccine from the sachet and subsequent uptake via the tonsils and mucous membrane in the oral cavity. It is clear that a bait must be attractive for dogs in terms of palatability. However, a highly attractive bait together with other bait variables, like size, shape and texture, can result in a failed vaccination attempt. If a bait is swallowed without sufficient chewing, the vaccine will not be released in the oral cavity and so cannot induce an immune response. Also, excessive spillage of the vaccine during bait handling by the animal can considerably reduce the amount of vaccine available for oral uptake. Finally, the integrity of the bait can also affect vaccination success. The bait matrix and sachet should not be easily separated [31,32]. For example, the fish meal bait used for the ORV of foxes in Europe is not only very well accepted by dogs, it also often falls apart during bait handling by the dog or when tossed to the dog. Subsequently, the dog will consume the bait matrix but leaves the sachet untouched [20,33]. Finally, external factors can also impact bait acceptance and handling by the dogs like the presence of other dogs or people, period of the day, and experience of the vaccinators, as identified in previous studies [20,21,22,33]. Therefore, selecting a bait that is well accepted and provides timely release in the oral cavity is imperative for effective ORV, but it cannot be guaranteed that all dogs accepting a bait will subsequently be vaccinated successfully. The selected egg-flavored bait and sachet have been shown to be readily accepted by dogs and to effectively release the vaccine in the oral cavity of the dogs [19,33,34]. 

Several (field-) experimental studies have already investigated the immunogenicity of different oral rabies vaccine constructs in dogs [35,36,37,38,39,40,41]. In Bali, a small experimental study with the second-generation oral rabies vaccine SAG2 showed that local dogs seroconverted after the consumption of a single bait [42]. The study presented in this paper is the first to test immunogenicity under local field conditions in Bali. Immunological studies in local (free-roaming) dogs are considered critical as their diet is of low quality and/or quantity, having a possible negative impact on the immune response [43,44]. Also, the presence of endo- and ectoparasites and other immune-compromising conditions can induce immune suppression [45]. Hence, these and other stress factors can impact seroconversion, including the duration and level of detectable antibodies after vaccination against rabies [46]. As in previous studies, SPBN GASGAS was shown here to be able to induce a detectable immune response comparable to that after parenteral vaccination [17]. 

Assessing the presence of antibodies against rabies virus is necessary in determining the immune status achieved by rabies vaccination. Several serological tests have been developed to detect antibodies against rabies virus. Detection of virus neutralizing antibodies by rapid fluorescent focus inhibition test (RFFIT) or fluorescent antibody virus neutralization (FAVN) is the gold standard. A live RABV is used in both tests, so these tests can only be performed in laboratories meeting high safety standards. The ELISA test does not require the use of a live virus, so it can be performed under less stringent safety conditions. Further, the immune response measured by the ELISA test used in this and other studies has been shown to be better suited than the results obtained with virus neutralizing antibody assays (FAVN and RFFIT) for predicting protection against rabies infection in orally vaccinated animals [19,24,25,47]. Mass dog vaccination campaigns including the use of ORV are required to generate herd immunity and the antibody level of individual dogs is less important; what matters is if the dog is protected against a rabies infection. 

The dogs orally vaccinated (either offered a bait or d.o.a) produced a detectable immune response. The seroconversion rate found in this study was higher than that found in Haitian and Namibian dogs that were also given the SPBN GASGAS vaccine and assessed using the same ELISA assay [16,18]. In a recent immunogenicity study with local Thai dogs kept in a dog shelter, 100% of the dogs vaccinated by the oral route had detectable levels of antibodies 28 dpv, again using the same vaccine and ELISA assay method. These dogs in Thailand, however, had received vaccinations against some common infectious diseases (canine distemper, parvovirus infection, adenovirus infection, bronchitis, and leptospirosis). They also received helminth treatment when they were 3 months old. These animals were fed on a daily basis with commercially available high-quality pet food and thus the dogs were in a very good condition [17]. 

It is unlikely that the physical condition of the dogs alone can account for the difference in seroconversion following oral vaccination with SPBN GASGAS between these different studies. It has been shown that the development of antibodies is slower after oral vaccination compared to parenteral vaccination [17]. In the Thai dog study, not all orally vaccinated dogs developed detectable antibodies (as assessed by the ELISA test) until 4 weeks post vaccination, whereas all parenterally vaccinated dogs had seroconverted by 7 dpv [17]. The short interval between vaccination and sampling (17–20 dpv) may explain the relatively low seroconversion rate for the Haiti study [16]. Another factor that likely affected the seroconversion rate in the latter study was the study design; the baseline blood sample was collected immediately after bait consumption [16]. Hence, the free-roaming dogs were surrounded by the vaccination team to prevent them from wandering off after bait consumption. This made the dogs anxious, possibly negatively influencing bait handling and consumption by the dog and consequently the vaccine virus’s release in the oral cavity, as indicated previously a pre-requisite for successful vaccination. 

Successful oral vaccination attempts are dependent not only on an efficacious vaccine and an attractive bait, but also on external factors such as the circumstances under which baits are offered to the dogs. An even more stressful situation occurred in the study in Namibia under which blood samples were collected several weeks prior to vaccination and for logistical reasons most animals were not offered a bait on their own premises, but the normally free-roaming dogs were brought to a central point for ORV baiting [18]. It is considered likely that the different environmental and stressful conditions may have had a negative effect on bait uptake by individual dogs. Many dogs were stressed as they were on a leash, to which they were not accustomed. They were in unfamiliar terrain and many unfamiliar dogs and humans were around them. The bait acceptance rate (61%) observed in the Namibia study was much lower compared to other studies with the same bait [20,22]. Suboptimal conditions can be expected to compromise bait uptake. Consequently, the vaccine is not released sufficiently or at all in the oral cavity and no immune response is induced after bait consumption. In contrast, a recent field study in Thailand showed that when free-roaming dogs were offered a bait directly, most animals accepted the bait readily and perforated the sachet [48]. It can be assumed that with careful management under real-field scenario conditions, effective bait-uptake can be optimized, resulting in high levels of seroconversion post vaccination, as has been observed in this study in Bali.

## 5. Conclusions

Bali island is a suitable target for achieving freedom from rabies, but it has been difficult to achieve the high levels of vaccination coverage required with its very large number of free-roaming dogs [8]. These free-roaming dogs are considered to play a key role in the transmission of rabies [12,49,50]. Efforts to eliminate rabies from Bali have received considerable international attention and, if successful, an enhanced Bali rabies elimination campaign with the use of ORV would set a precedent for the elimination of dog-mediated rabies elsewhere. In many countries, traditional parenteral methods have failed to reach adequate vaccination coverages. ORV can be very helpful in achieving adequate levels of herd immunity [12]. 

This study confirmed that local dogs in Bali, like local dogs in other parts of the world, develop an adequate immune response after a single oral vaccination with the third-generation oral rabies vaccine strain SPBN GASGAS. These results underline the potential of ORV as an important tool for targeting hard-to-reach free-roaming dogs in mass dog vaccination campaigns in Bali. The next step should be the implementation of field trials integrating ORV into mass dog vaccination strategies at a large scale and investigating the cost-effectiveness of this approach in Bali.

## Figures and Tables

**Figure 1 viruses-15-01405-f001:**
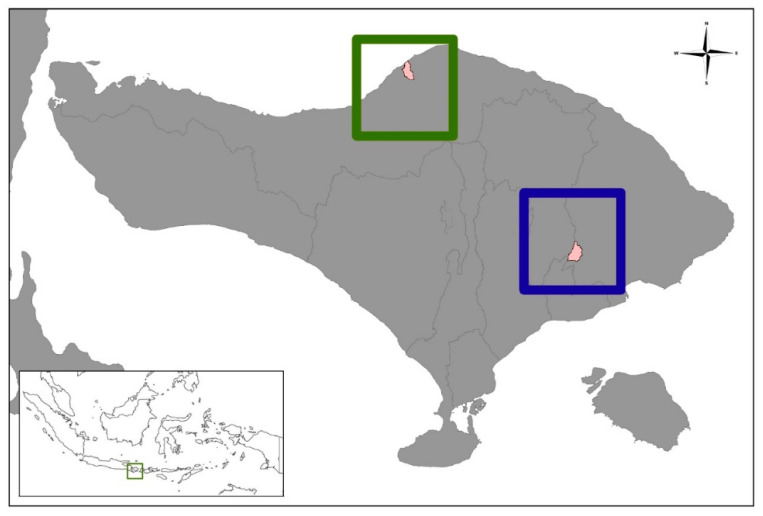
The serology investigations were conducted in Nongan village, Karangasem (blue-square) and Banyuning village, Buleleng (green square) on Bali Island.

**Figure 2 viruses-15-01405-f002:**
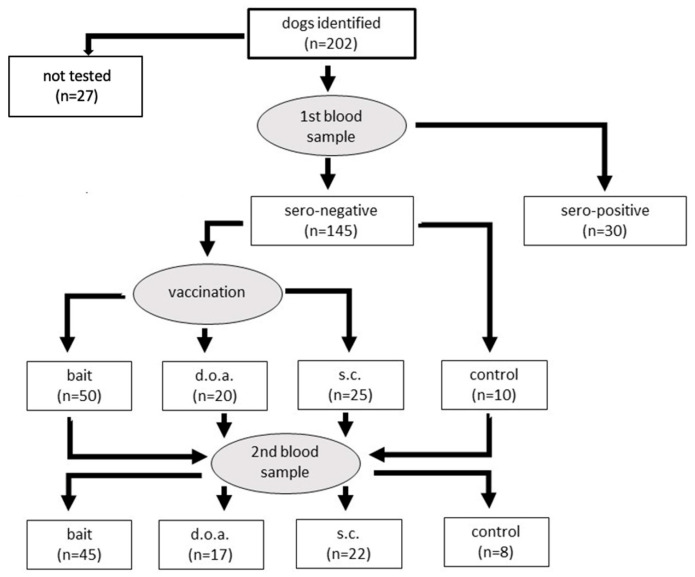
Flow chart of the study and number of dogs per group.

**Figure 3 viruses-15-01405-f003:**
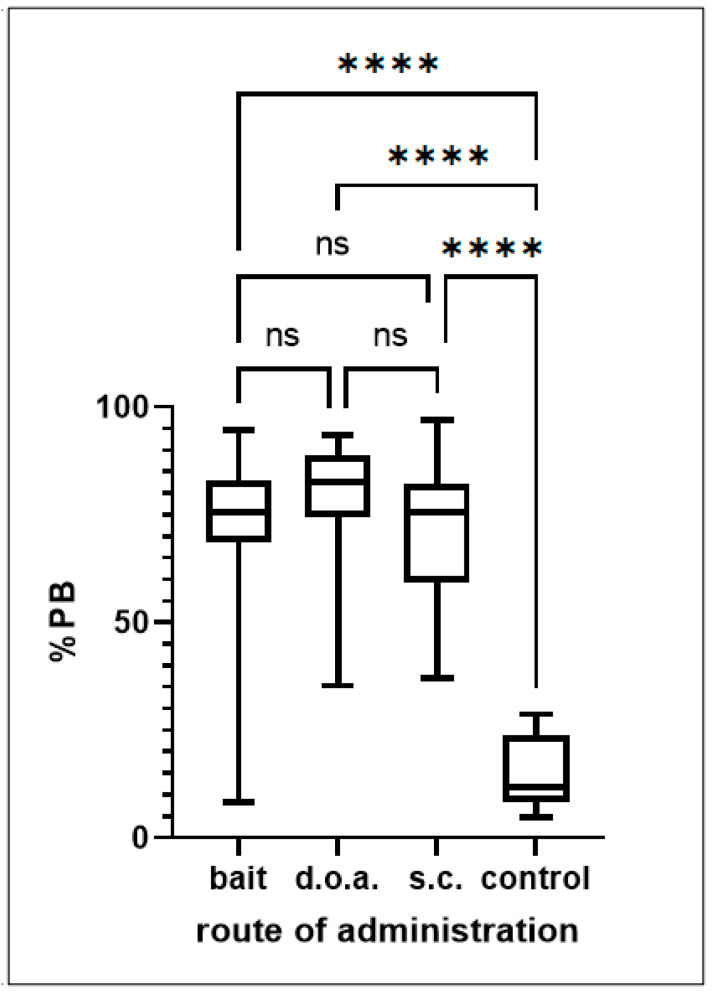
Box plot of the measured immune response post vaccination (ELISA-%PB) of the four different treatment groups; blood samples were collected between 27 and 32 dpv (ns—not significant, *p* < 0.05; ****—highly significant, *p* < 0.0001).

**Table 1 viruses-15-01405-t001:** Seroconversion rate of dogs in the different treatment groups for each independent variable based on ELISA results (cut-off: ≥40% PB). (n—number of animals that tested seropositive; N—number of animals tested).

Variable	Treatment								Total	
	Bait		d.o.a.		Parenteral		Control		(excl. Control)	
Area	n/N	%	n/N	%	n/N	%	n/N	%	n/N	%
- Karangasem	26/29	89.7	10/11	90.9	16/18	88.9	0/4	0	52/58	89.7
- Buleleng	14/16	87.5	6/6	100	4/4	100	0/4	0	24/26	92.3
Size										
- Small	18/22	81.8	9/10	90.0	12/13	92.3	0/0	0	39/45	86.7
- Medium	22/23	95.7	7/7	100	8/9	88.9	0/7	0	37/39	94.9
- Large	-	-	-	-	-	-	0/1	0	-	-
Sex										
- Male	24/26	92.3	11/12	91.7	11/13	84.6	0/6	0	46/51	90.2
- Female	16/19	84.2	5/5	100	9/9	100	0/2	0	30/33	90.9
Age										
- Juvenile	20/24	83.3	9/10	90.0	9/9	100	0/3	0	38/43	88.4
- Adult	20/24	83.3	7/7	100	11/13	84.6	0/5	0	38/44	86.4
Supervision										
- Restricted	13/14	92.9	6/6	100	8/9	88.9	0/3	0	27/29	93.1
- Free-roaming	27/31	87.1	10/11	90.9	12/13	92.3	0/5	0	49/55	89.1
Total	40/45	88.9	16/17	94.1	20/22	90.9	0/8	0	76/84	90.5

## Data Availability

The original data can be provided upon reasonable request and should be directed to the corresponding author.

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
