# Peer review of "Immunogenicity of Oral Rabies Vaccine Strain SPBN GASGAS in Local Dogs in Bali, Indonesia"

_viruses, 2023, doi:10.3390/v15061405_

Round 1
Reviewer 1 Report
This paper describes a rabies vaccination study in dogs in Bali using ORV in baits. The study was appropriately designed and the manuscript fairly well written. Specific recommendations to improve clarity of the paper were included directly on the manuscript. The authors should be careful to use "rabies" only when speaking of disease and to use "rabies virus" in all other circumstances throughout the paper. Please discuss how the ELISA compares to other serologic tests for antibodies to rabies virus, especially with respect to what is considered a protective response against the disease.

Author Response
Reviewer 1
General comment:
This paper describes a rabies vaccination study in dogs in Bali using ORV in baits. The study was appropriately designed and the manuscript fairly well written. Specific recommendations to improve clarity of the paper were included directly on the manuscript. The authors should be careful to use "rabies" only when speaking of disease and to use "rabies virus" in all other circumstances throughout the paper. Please discuss how the ELISA compares to other serologic tests for antibodies to rabies virus, especially with respect to what is considered a protective response against the disease.
Reply: we carefully checked the manuscript to make sure that we clearly distinguish ‘rabies’ (the disease) and ‘rabies virus’ (the causative agent). Furthermore, we discussed the issue with the ELISA and the threshold indicative of protective immune response in more detail.
Specific comments:
Line 93: insert “prior to vaccination”
Reply: has been inserted
Line 99: remove extra comma
Reply: comma has been removed
Line 106: provide a better explanation how the “health” of the dogs were monitored. How were the dogs specifically examined for adverse effects?
Reply: the following has been added: “by visual examination during house visits”
Line 150: add “at each study site”?
Reply: Thanks for pointing this out. At this stage of the document, we are still describing the overall protocol so we prefer to change “25” to “50” instead of adding the suggested change. We added in line 153 “to 45”
Line 155: I assume you mean ‘canine parvovirus’ – write out.
Reply: we change “parvo” to “canine parvovirus infection”
Line 175: delete “Also, here”
Reply: corrected as suggested
Line 176: insert “with the comparison also”
Reply: We prefer to correct this sentence in another way, as the suggested change is considered confusing.
Line 183: change “has been” to “was”
Reply: corrected as suggested
Line 220: delete “sample” (2x)
Reply: corrected as suggested
Line 222: delete “one”
Reply: corrected as suggested
Line 226: delete “The fact …. sero-converted” and change to “The lack of seroconversion in some of the vaccinated animals”
Reply: corrected as suggested
Line 227: change “vaccine efficacy” to “vaccination”
Reply: corrected as suggested
Line 233: delete “time” and “the”
Reply: corrected as suggested
Line 241: delete “the”
Reply: corrected as suggested
Line 242: change “methods” to “serological tests”
Reply: corrected as suggested
Line 243: change “rabies” to “rabies virus”
Reply: corrected as suggested
Line 246: change “carried out” to “performed”
Reply: corrected as suggested
Line 249-257: These paragraphs seem out of order. Combine the two and move to Conclusions
Reply: We agree with the Reviewer and this paragraph (in the original version it was a single paragraph) has been move to the Conclusion.
Line 249: change “freedom from rabies” to “rabies-free status”
Reply: corrected as suggested
Line 250: delete “shown”
Reply: corrected as suggested
Line 250: change “vaccinating” to “to vaccinate”
Reply: corrected as suggested
Line 256: change “oral rabies vaccine” to “ORV”
Reply: Unfortunately, the abbreviation ORV is often used for both ‘oral rabies vaccine’ and ‘oral rabies vaccination’. The tripartite (WHO/WOAH/FAO) is using the abbreviation for oral rabies vaccination, we prefer to stick with this and thus we have not made the suggested change.
Line 260: change “as what was” to “than that”
Reply: corrected as suggested
Line 261: delete “and”
Reply: corrected as suggested
Line 269: delete “However,”
Reply: corrected as suggested
Line 281: insert “possibly”
Reply: corrected as suggested
Line 283: delete “Therefore, essential for a”
Reply: corrected as suggested
Line 283: change “attempt to “attempts”
Reply: corrected as suggested
Line 284: insert “dependent on”
Reply: corrected as suggested
Line 287: change “but” to “Instead” and start a new sentence
Reply: corrected as suggested
Line 290: change “bait consumption (handling) of” to “bait uptake by”
Reply: corrected as suggested
Line 296/297: change “However” to “In contrast”
Reply: corrected as suggested
Reviewer 2 Report
This is a well presented study confirming the immunogenicity of an oral rabies vaccine in field conditions. The methods used are clear and rational. The possible explanations for failure to achieve 100% sesroconversion are excellent.
1 How was the vaccine bait administered? Were the all the dogs observed to have consumed the bait? What happened if the dog refused the bait?
2 Is the vaccine sufficiently attenuated to be distributed in rural or urban areas, aimed at stray dogs?
3 Is there any evidence that the dogs which did not seroconvert had a booster response on vaccination a year or more later?
4 Typo?
259 The seroconversion rate we found in our study was higher as what was found in Haitian and Namibian dogs ….
Should this be: was higher than that what was found….
5 293 In fact, the bait acceptance rate (61%) observed in this study was much lower than compared to other studies with the same bait
does ‘this’ refer to the Namibia study or the Bali study?
Suggest clarification.
typos described above
Author Response
Reviewer 2
Suggestions for Authors
This is a well presented study confirming the immunogenicity of an oral rabies vaccine in field conditions. The methods used are clear and rational. The possible explanations for failure to achieve 100% sesroconversion are excellent.
Comment 1: How was the vaccine bait administered? Were the all the dogs observed to have consumed the bait? What happened if the dog refused the bait?
Reply: We added some information on bait offering and – consumption in the M&M section. Also, here it was described what was done when an animal did not accept the bait.
Comment 2: Is the vaccine sufficiently attenuated to be distributed in rural or urban areas, aimed at stray dogs?
Reply: yes, the vaccine strain meets the international (WOAH) safety requirements as specified in the text.
Comment 3: Is there any evidence that the dogs which did not seroconvert had a booster response on vaccination a year or more later?
Reply: This was not part of this study. However, previously it was shown that if an animal did not seroconvert (using the ELISA) after 4-6 weeks post vaccination, it would be considered a vaccination failure as the animal would not be protected against a subsequent challenge infection. Recent results from a study in Thailand showed that a booster effect was detectable after 3 years post vaccination. These results will be presented in a separate paper. Based on the comments of the editorial office and other reviewers, the issue with immune response detected by ELISA and its indicative value for protection is discussed in more detail in the revised manuscript
Comment 4: Line 259 -Typo error? - The seroconversion rate we found in our study was higher as what was found in Haitian and Namibian dogs …. Should this be: was higher than that what was found….
Reply: has been corrected
Comment 5: Line 293 - In fact, the bait acceptance rate (61%) observed in this study was much lower than compared to other studies with the same bait does ‘this’ refer to the Namibia study or the Bali study? Suggest clarification.
Reply: It refers to the Namibian study and this has been clarified in the text
Round 2
Reviewer 1 Report
Discussion still needs some improvement in word choice and minor editing, as noted directly on the manuscript. In particular, the discussion about individual dog titers being irrelevant should be modified. While they are not as important as herd immunity, individual dog titers are related to protection against rabies and are a measure of such.

Author Response
Reviewer 1
General comments
Discussion still needs some improvement in word choice and minor editing, as noted directly on the manuscript. In particular, the discussion about individual dog titers being irrelevant should be modified. While they are not as important as herd immunity, individual dog titers are related to protection against rabies and are a measure of such.
Reply: the section on individual immune response has been adapted taking the comments and suggested changes of the reviewer in consideration. Also, the whole manuscript has been edited by a native speaker to improve the document
Specific comments
Line 240: insert ‘as’ before ‘many’
Reply: inserted
Line 247: insert a ‘comma’ (twice)
Reply: inserted twice
Line 250: insert ‘by the animal’
Reply: inserted
Line 257: delete hyphen
reply: deleted
Line 259: replace ‘guaranteeing’ by ‘provides’
Reply: replaced accordingly
Line 260: replace ‘circumvented’ by ‘guaranteed’
Reply: replaced accordingly
Line 262: replace ‘subsequently’ by ‘to’
Reply: replaced accordingly
Line 278: delete ‘for serological test’
Reply: deleted
Line 281: change ‘performed’ to ‘performed’
Reply: corrected
Line 286: write out full word (incl.)
Reply: adapted accordingly
Line 286-289: several comments and suggestions were made
Reply: Sentences have been corrected accordingly
Line 316: delete ‘essential for’
reply: deleted
Reviewer 2 Report
The amendments improve the paper
The changes are good but the English grammar needs editing and improving for these German speakers.
Author Response
Reviewer 2
The changes are good but the English grammar needs editing and improving for these German speakers.
Reply: manuscript has been checked by native English speaker and edited accordingly